# Isolation and Structural Characterization of Alginates from the Kelp Species *Laminaria ochroleuca* and *Saccorhiza polyschides* from the Atlantic Coast of Morocco

Soukaina Kaidi [1] , Fouad Bentiss [2,3], Charafeddine Jama [3] , Khaoula Khaya [1], Zahira Belattmania [1] , Abdeltif Reani [1] and Brahim Sabour [1,*]

[1] Phycology, Blue Biodiversity & Biotechnology R.U., Laboratory of Plant Biotechnology, Ecology and Ecosystem Valorization, URL–CNRST n°10, Faculty of Sciences, University Chouaib Doukkali, P.O. Box 20, El Jadida M-24000, Morocco

[2] Laboratory of Catalysis and Corrosion of Materials, Faculty of Sciences, University Chouaib Doukkali, P.O. Box 20, El Jadida M-24000, Morocco

[3] Materials and Transformations Unit, University of Lille, CNRS, INRAE, Centrale Lille, UMR 8207-UMET, F-59000 Lille, France

* Correspondence: sabour.b@ucd.ac.ma

**Abstract:** Alginates are linear unbranched polysaccharides produced by brown seaweeds. The gel properties of this phycocolloid are mainly linked to the monomer composition, sequential structure and molecular weight of the polymeric chain. Nevertheless, these structural parameters are also dependent on the age and tissue of the thalli used. In this work, the kelp species *Laminaria ochroleuca* and *Saccorhiza polyschides* harvested from the Moroccan coast were analyzed for their alginates content and composition in different thalli parts (blade and stipe). Spectroscopic characterization ($^1$H NMR and FT-IR), viscosity, and molecular weight of the extracted alginates were investigated. The obtained results showed that the alginate contents ranged from $20.19 \pm 2$ to $49.8 \pm 2.4\%$ dw. FT-IR spectroscopy exhibited similar spectra between the alginates extracted from both species and the commercial alginate. The $^1$H-NMR results revealed interesting variations in block composition between species, and less disparity with regard to the tissue type (M/G values ranging from 0.89 to 2.14). High mannuronate content was found in all alginates except for those extracted from the blade of *L. ochroleuca*. The homopolymeric fractions $F_{MM}$ are remarkably high compared to the $F_{GG}$ and heteropolymeric fractions ($F_{GM}/F_{MG}$) in *S. polyschides*. However, for *L. ochroleuca*, the heteropolymeric fractions are quite abundant, accounting for more than 34% of the polymer diads. The alginates extracted from both species indicated low values for the intrinsic viscosity. Based on the yield and the structural properties of their alginates, this study suggests that both *L. ochroleuca* and *S.polyschides* could be considered potential alginophytes to be incorporated into the industry of alginates. It also proposes an optimization of the yield, using the different thalli parts in the extraction (blade and stipe). The chemical structure and viscosity of their alginates may contribute to expanding their applications, especially in biotechnology and medical fields.

**Keywords:** kelps; *Laminaria ochroleuca*; *Saccorhiza polyschides*; alginates; $^1$H NMR; FT-IR; viscosity

## 1. Introduction

Alginates being some of the main seaweeds' biomolecules, they were found to have impressive assets such as various bioactivities, biocompatibility, and biodegradability, as well as their emulsifying and chelating characters. These distinctive biochemical features along the gelling, viscosifying, and stabilizing properties are the main reasons behind the hydrocolloid diversified applications [1]. In fact, alginates are commonly used in food and feed, agricultural, textile, pharmaceutical, and cosmetic industries, and were lately found to have interesting applications in bioscience and bioengineering fields. However,

the food and textile remain the most prominent markets for alginates along with the foremost expanded and growing cosmetics business [2,3]. In molecular terms, alginates are family of unbranched binary linear copolymers of (1-4)-linked β-D-mannuronic acid (M) and α-L-guluronic acid (G) monomers that form varying block structures comprising homogeneous blocks of consecutive M residues (MM), G-block sequences based entirely on L-guluronic acid (GG), and alternating units of M and G residues (GM/MG) [4,5].

The actual physical properties of alginate and its resultant hydrogels were found to be mainly affected by the variation in M and G contents (M/G ratio), as well as the length of G-block and the molecular weight. These characteristics are known to significantly control the stability and viscosity of alginate hydrocolloids. Essentially, the high gel strength and stiffness are proportional to a high concentration of homopolymeric G-blocks. The latter is involved along with the quantity of the external salt, acidic conditions, as well as the time and temperature of the extraction in intermolecular cross-linking reaction with divalent cations for the elaboration of hydrogels [2]. In parallel with the mentioned aspects, the source of alginates is also a major criterion affecting the composition of alginates which differs considerably within the seasons, and algal tissues [6–8]. This variation is thought to reflect the functional role of the polysaccharide considered as a structural component in the cell walls of marine brown algae (Phaeophyceae) that provides adequate flexibility to their thalli [9]. In order to cover the increasing demands for alginate production, kelps and fucoids are harvested from natural populations and seaweed farming activities to produce around 200 different alginates commercially available [10,11]. *Laminaria ochroleuca* and *Saccorhiza polyschides* are two kelp species that occur along the Moroccan Atlantic coast. The tridimensional structure of their thalli offers interesting physicochemical properties to be potentially applied in alginate extraction through using the species in all its integrity.

The present study explores the fine structure of alginates extracted from different tissues (blade and stipe) within thalli of *L. ochroleuca* and *S. polyschides*. The exploitation of the stipe and the blade would improve the yield of the extracted sodium alginates while providing more chemical structures that may expand the hydrocolloid applications. Both kelps harvested during spring from the Atlantic coast of Morocco were investigated for content, spectroscopic characterization ($^1$H NMR and FT-IR) as well as viscosity of their extracted sodium alginates.

## 2. Materials and Methods

### 2.1. Sampling and Biomass Preparation

*S. polyschides* and *L. ochroleuca* (Table 1) were collected at low tide in May 2018, on a moderately wave-exposed rocky shore of El Jadida (33°24′60.7″ N–8°55′68.5″ W) along the northwestern Atlantic coast of Morocco. Both kelps were hand-picked from the intertidal rocky substratum using a metal scraper to preserve the entire thalli structure. In the laboratory, the harvested samples were washed with freshwater to remove impurities and epiphytes. This step was followed by a second cleaning with distilled water after which the thalli were cut to distinguish parts namely the blade and the stipe to be used separately for alginates extraction. Seaweeds samples were air-dried for 7 days, and then put in a ventilated oven at 45 °C to constant dry weight. Ash content in the algal biomass was determined by incineration of samples at 450 °C for 3 h in the muffle furnace B180 (Nabertherm GmbH, Lilienthal, Germany).

**Table 1.** Characteristics of *L. ochroleuca* and *S. polyschides* during May 2018.

| Parameters | | *L. ochroleuca* | *S. polyschides* |
|---|---|---|---|
| Length (cm) | (blade) | 61.8 ± 2.8 | 76.18 ± 10 |
| | (stipe) | 6.92 ± 0.5 | 3.16 ± 1.8 |
| Fresh Weight (g) | (blade) | 227.6 ± 23.3 | 289.66 ± 85 |
| | (stipe) | 6.75 ± 1 | 5.22 ± 7 |

**Table 1.** *Cont.*

| Parameters | | *L. ochroleuca* | *S. polyschides* |
|---|---|---|---|
| Moisture (%) | (blade) | 5.7 ± 4 | 7.9 ± 0.6 |
| | (stipe) | 4.37 ± 1.4 | 5 ± 0.9 |
| Ash content (%) | (blade) | 0.50 ± 1.2 | 0.98 ± 2.4 |
| Density (2 m$^2$) | | 3.0 ± 1.0 | 3.0 ± 3.0 |
| Fertility (%) | | 64.1 ± 5 | 0 |

### 2.2. Extraction and Purification of Alginates

Separate extractions were performed in triplicate according to the Calumpong et al. [12] modified procedure. First, 12.5 g of dried samples belonging to each thalli part were put separately as small pieces in 2% formalin solution during 24 h. After the hydration treatment, samples were washed with distilled water and soaked in 0.2 N HCl solution for 24 h. Following this, samples were again washed with distilled water, before being placed in 2% sodium carbonate solution during 24 h for the extraction. The obtained soluble fraction was filtered and centrifuged (4500× *g*, 20 min) to collect the filtrate, while the remaining solid residue was used to repeat the procedure. Ethanol 95% and acetone 90% were used for the precipitation and purification of sodium alginates. The polysaccharide extract was then placed in an oven at 45 °C for drying constant weight. Alginates yields are estimated as a percentage of the initial dry weight of the seaweed (% dw).

### 2.3. FT-IR Spectroscopy Analysis

Infrared spectra of the commercial sodium alginate (CAS No. 9005-38-3, Lot MKBQ4519V, Sigma-Aldrich, Gillingham, UK) and the alginates extracted from the stipe and the blade of both kelps were recorded at room temperature in the 400 to 4000 cm$^{-1}$ range using Nicolet Impact 400D FT-IR Spectrometer (Nicolet Impact, Madison, WI, USA). A total of 64 scans were averaged for each sample at 4 cm$^{-1}$ resolution. The OMNIC 7.1 software (Nicolet, Madison, WI, USA) was used for the processing of the infrared spectra plots.

### 2.4. Proton Nuclear Magnetic Resonance Spectroscopy ($^1$H NMR)

To prepare samples for 1H NMR analyses, Na alginates were dissolved in $D_2O$ and dried several times prior to NMR spectrum acquisition. The 1H spectra were achieved at 9.4T (Proton Larmor frequency of 400.33 MHz) on the spectrometer AV II 400 MHz (Bruker Corporation, Billerica, MA, USA) using a 5 mm Triple Resonance Broadband Inverse probe, and with temperature regulated at 343 K. Each spectrum consisted of 16 K size of Free Induction Decay recorded with a sweep width of 4800 Hz. Presaturation was applied during the relaxation delay and mixing time. The raw data were apodized by 0.5 Hz of exponential line broadening prior to Fourier transformation. A total of 32 scans were acquired.

### 2.5. Intrinsic Viscosity [η] and Molecular Weight (Mv)

Viscosity measurements for alginate samples were performed at 25 °C using Ubbelohde viscometer with a 0.5 mm capillary diameter. The flow time of the solutions was measured relative to T0 which corresponds to the solvent flow time (0.1 M NaCl). Measurements were performed over a range of polysaccharide concentrations from 0.05 to 0.5 g/dL. Three measurements were performed for each solution and the resulting averages were used to calculate the relative viscosities. The intrinsic viscosity [η] is defined by the Huggins equation where $\eta_{sp}$, C, and $K_H$ are the specific viscosity, the concentration of the solution, and the Huggins constant, respectively.

$$\eta_{sp}/C = [\eta] + K_H[\eta]^2 C$$

The average molecular weight (Mv) was calculated based on the intrinsic viscosity using the Mark–Houwink–Sakurada equation: $[\eta] = k(M_v)^a$ where a and k are empirical parameters that depend on the temperature, the ionic strength, and the alginate composition. The values k and a are used to determine the average molecular weight (Mw) are 0.023 and 0.984, respectively, as proposed by Torres et al. [13].

## 3. Results and Discussion

### 3.1. Alginate Yield

Alginate contents were determined for the stipe and the blade of both *L. ochroleuca* and *S. polyschides*. The alginate contents values ranged from 20% dw to 49% dw in different thalli parts from both species (Table 2). The highest alginate content (49.8 $\pm$ 2.4% dw) was obtained from the stipes of *L. ochroleuca*. The lower alginate content was recorded in the *S. polyschides* blades (20.19 $\pm$ 2 dw). These results are in agreement with the ranges reported from other kelps also known as worldwide alginophytes such as: *Laminaria hyperborea* [6], *Macrocystis pyrifera* [14], *Durvillaea potatorum* [15], *Saccharina japonica* [16], *Laminaria digitata* [17], and *Ascophyllum nodosum* [18]. According to several studies, the highest alginate content is usually found in blades [5,19]. This is not the case for the present work since the highest yield was obtained from the stipe. This disparity might be explained by the seasonal trends in alginate content, in addition to the species' own characteristics [6,14]. Similar results were obtained from other kelps such as *D. antarctica* [19].

**Table 2.** Alginate yields of *L. ochroleuca* and *S. polyschides* (blades and stipes) compared to various kelp species.

| Kelp Species | Alginates Yield | References |
|:---:|:---:|:---:|
| *Macrocystis pyrifera* | 18–45% | [14] |
| *Laminaria digitata* | 16–36% | [17] |
| *Laminaria hyperborea* | 14–21% | [6] |
| *Saccharina japonica* | 17–25% | [16] |
| *Saccharina latissima* | 16–34% | [17] |
| *Lessonia trabeculata* | 13–29% | [20] |
| *Durvillaea potatorum* | 55% | [15] |
| *Ascophyllum nodosum* | 12–16% | [18] |
| *Laminaria ochroleuca* (blade) | 28.96 $\pm$ 0.20 | |
| *Laminaria ochroleuca* (stipe) | 49.80 $\pm$ 2.40 | This study |
| *Saccorhiza polyschides* (blade) | 20.19 $\pm$ 2.00 | |
| *Saccorhiza polyschides* (stipe) | 26.28 $\pm$ 0.08 | |

### 3.2. FT-IR Spectroscopy

The FT-IR spectra of the commercial alginate (Sigma-Aldrich Na-Alginate), and sodium alginates extracted from *L. ochroleuca* and *S. polyschides* (blade and stipe) are presented in the Figure 1. Despite the difference in bands intensity, all spectra demonstrated a very marked resemblance, as they showed similar positions of the characteristic bands between sodium alginate extracted from *L. ochroleuca*, *S. polyschides* and the commercial alginates. The similarity was also manifested between tissues from the same thalli as they revealed similar spectra of alginates extracted from the two structural parts (blade and stipe) of each species. In the 3600–1600 $cm^{-1}$ region, all spectra showed broad peaks centered in 3453 $cm^{-1}$ and 3438 $cm^{-1}$ which could be assigned to hydrogen bonded O–H stretching vibrations. The weak signals at 3016 $cm^{-1}$ in the blade spectra, and at 3033 $cm^{-1}$ and 2937 $cm^{-1}$ in the stipe spectra of *L. ochroleuca* and *S. polyschides*, respectively, were attributed to C–H stretching vibrations. Asymmetric stretching of carboxylate O–C–O

vibrations were noted around 1600 cm$^{-1}$ in all spectra, which reveals a similar structure between all the extracted alginates. However, slight differences between spectra were found at the region 1200–1500 cm$^{-1}$ which is considered as one of the richest regions in structural information [21]. Strong peaks appeared at 1433 cm$^{-1}$ exclusively in the blade spectra of both species, referring to symmetric stretching vibrations of carboxylate groups. The bands at 1404 cm$^{-1}$ were attributed to C–OH deformation vibration with contribution of O–C–O symmetric stretching of carboxylate O–C–O vibrations [22]. These peaks are frequent in sodium alginate with an M/G ratio above 1, referring to a high content in the mannuronic acid units [19,23]. The extracted alginates exhibited bands at the range of 1023–1026 cm$^{-1}$ in all spectra due to the C-O group [24]. Moreover, the FT-IR analysis revealed the absence of peaks around 1230–1280 cm$^{-1}$ related to the sulphate ester groups (S=O) occurring in sulphated polysaccharides (such as fucoidan or laminarin). The lack of such a band in all spectra reflects their highly purified sodium alginates. The anomeric region (950–750 cm$^{-1}$), also called the fingerprint, is the most discussed in carbohydrates [25]. All spectra exhibited weak bands between 928 cm$^{-1}$ and 930 cm$^{-1}$ assigned to the C–O stretching vibration of uronic acid residues, in addition to peaks between 847 cm$^{-1}$ and 860 cm$^{-1}$ attributed to the C1–H deformation vibration of β-mannuronic acid residues.

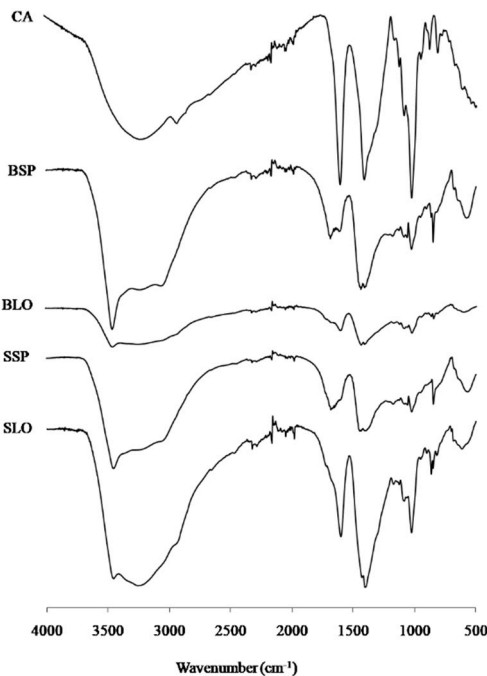

**Figure 1.** FT-IR spectra of commercial sodium alginate (CA) and alginates extracted from the blades *L. ochroleuca* (BLO), stipes of *L. ochroleuca* (SLO), blades of *S. polyshides* (BSP), and stipes of *S. polyshides* (SPS).

### 3.3. $^1$H NMR Spectroscopy Analysis

$^1$H NMR spectroscopy is a method that has consistently been used for the complete characterization of alginates, and much of their detailed block structure [26]. All studied alginates exhibited typical 400 MHz-1H NMR with three key signals (Figure 2) attributed to the anomeric hydrogen of guluronic acid (G) at 5.1–5.2 ppm (pic I), the anomeric hydrogens of mannuronic acid (M1), and the H-5 of alternating blocks (GM-5) overlapping at 4.7–4.9 ppm (pic II), in addition to the H-5 of guluronic acid residues in the homopolymeric G blocks, at 4.5–4.6 ppm (pic III). Determination of the molar pair frequencies of the monads (FG, FM) and the diad sequences (FGG, FMM, FMG, FGM), as well as the M/G ratio, and η parameter were calculated following the equations given by Grasdalen et al. [27]:

$$F_G = AI/(AII + AIII)$$

$$F_M = 1 - F_G$$

$$F_{GG} = AIII/(AII + AIII)$$

$$F_{GM} = FM_G = F_G - F_{GG}$$

$$F_{MM} = F_M - F_{MG}$$

$$M/G = (1 - F_G)/F_G$$

$$\eta = F_{MG}/(F_M - F_G)$$

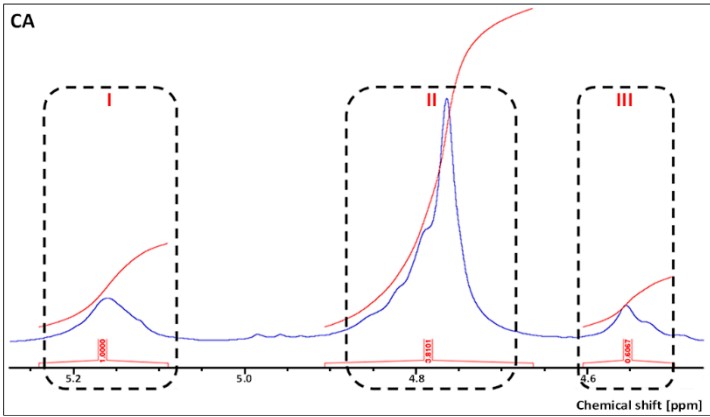

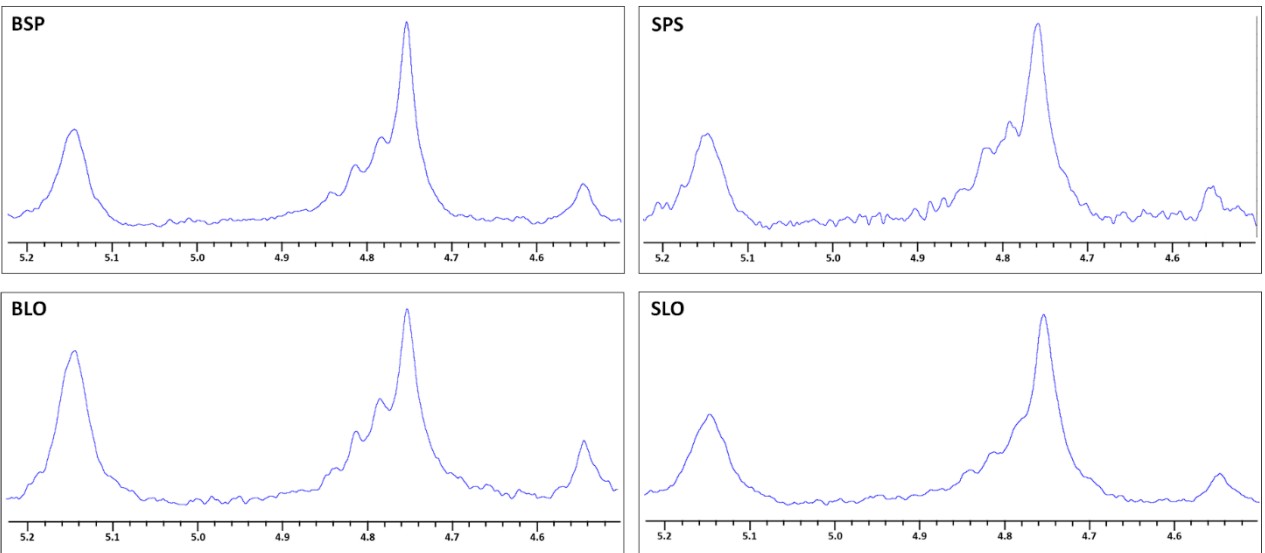

**Figure 2.** $^1$H NMR spectra of commercial sodium alginate (CA) and alginates extracted from the blades of *S. polyshides* (BSP), stipes of *S. polyshides* (SPS), blades *L. ochroleuca* (BLO), and stipes of *L. ochroleuca* (SLO).

The obtained M/G values were 1.38 from *L. ochroleuca'* stipe, 1.62 and 2.14 from *S. polyschides* blade and stipe, respectively. Whereas, the analyzed commercial alginate exhibited an M/G value exceeding 3.4 (Table 3). Such dominance was found in alginates extracted from various kelp species such as *Ascophyllum nodosum* [28], *Laminaria digitata* [29], *Durvillaea* species including *D. antarctica* [30], *D. potatorum* [31], *D. willana* [26], in addition to *Macrocystis pyrifera* [32] and *Saccharina japonica* [33], both known as some of the main sources of alginates in the phycocolloid industry. The M/G ratio, however, is not the only factor that controls the alginate gelling properties; they, indeed, depend fundamentally on the molar frequencies of pairs of uronic acids, namely, the homopolymeric block structures ($F_{MM}$, $F_{GG}$) and alternating blocks ($F_{MG}/G_M$). These structural parameters were found

to strongly affect the chemical and physical properties of alginate, since the stiffness of the chain blocks increases in the order: MM block < MG block < GG block. Therefore, it allows the classification of alginates as being: high-M ($F_M$ > 0.7), low-M ($F_M$ < 0.6), and intermediate-M ($F_M$, 0.6–0.7) [26]. Several studies have revealed the impact of the alginate's composition on its rigidity which was proven to be proportional to the content and length of the G-blocks. The former contributes to the physical strength of alginates through binding with sodium ions cooperatively and forming junctions in the gel network, unlike the MM-blocks and MG-blocks that comprise the gel network between the junctions, allowing for more flexibility to the seaweed thalli [19,34]. Analysis of alginates from the studied kelps species *L. ochroleuca* and *S. polyschides* showed interesting variations in block composition between species, and less disparity with regard to tissue type of each alga separately (Table 3). The alginate extracted from both parts of *S. polyschides* (blade and stipe) has essentially the same structure as typical commercial alginate, being classified as an intermediate type of alginate ($F_{MM}$ 0.6–0.7) following the $^1$H NMR results (Table 3). The latter indicates higher values of the homopolymeric regions ($F_{MM}$) in the commercial alginate ($F_{MM}$ = 0.7) compared to those found in the blade ($F_{MM}$ = 0.62) and the stipe ($F_{MM}$ = 0.68) of *S. polyschides*, although they were never less than 0.6. The homopolymeric blocks ($F_{MM}$) in the aforementioned samples were higher than the heteropolymeric fractions ($F_{GM/MG}$), which, in turn, were higher than the guluronic blocks ($F_{GG}$). A similar composition has previously been described for alginates from other kelps such as *Macrocystis pyrifera*, *Ecklonia arborea* [32], *Laminaria digitata* [27], *Saccharina japonica* [33,34], *Lessonia nigrescens* [30], *Ecklonia maxima* [1], *Durvillaea potatorum* [31], *Marginariella boryana* [26], and *Ascophyllum nodosum* [28]. In parallel, alginates extracted from the stipe and blade of *L. ochroleuca* yielded a low-M type of alginate ($F_M$ < 0.6) which contained a higher heteropolymeric fraction ($F_{GM/MG}$) compared to both homopolymeric blocks ($F_{MM}$) and ($F_{GG}$). The predominance of heteropolymeric sequences ($F_{GM/FMG}$) was also noticed in alginates extracted from *Saccharina longicruris* [35] and *Sargassum muticum* [8]. Nonetheless, the sequence distribution is mostly described by the η parameter, as η value < 1 reflects the abundance of homopolymeric blocks MM and GG, whereas η = 1 reveals completely random cases in algal alginates, and 1 < η < 2 illustrates the alternate-like cases MG and GM [27,35]. Clearly, all obtained η values of the evaluated samples exceeded one. These results reflect the dominance of the heteropolymeric fractions MG/GM over the homopolymeric fractions in all the tested alginates. Although those extracted from *S. polyschides* blade and stipe are slightly close to one, which might also reflect a random composition in their alginates structure and composition. According to Panikkar and Brasch [26], these samples, being intermediate alginates, have a poor correlation between their composition and estimates of their block length. Therefore, it might explain the discrepancy between the conclusion related to the η parameter and the composition exhibited by the $^1$H NMR spectroscopy analysis of the studied samples. The obtained η values are similar to those previously reported from alginates of other kelps such as *Saccharina longicruris* [35], *Ecklonia maxima* [1], and *Durvillaea willana* [26].

**Table 3.** Composition data of alginates from the blades and stipes of the studied kelps *L. ochroleuca* and *S. polyshides* in comparison with the main kelp species.

| Kelp Species | Composition | | | Sequence | | | | References |
|:---:|:---:|:---:|:---:|:---:|:---:|:---:|:---:|:---:|
| | $F_M$ | $F_G$ | M/G | $F_{MM}$ | $F_{GG}$ | $F_{MG, GM}$ | η | |
| *Macrocystis pyrifera* | 0.62 | 0.38 | 1.63 | 0.42 | 0.18 | 0.40 | 0.85 | [32] |
| *Laminaria hyperborean* (blade) | 0.45 | 0.55 | 0.81 | 0.28 | 0.38 | 0.34 | 0.69 | [29] |
| *Laminaria hyperborean* (stipe) | 0.32 | 0.68 | 0.47 | 0.2 | 0.56 | 0.24 | 0.55 | [29] |
| *Laminaria digitata* | 0.61 | 0.39 | 1.56 | 0.41 | 0.15 | 0.30 | 0.63 | [27] |

**Table 3.** *Cont.*

| Kelp Species | Composition | | | Sequence | | | | References |
|---|---|---|---|---|---|---|---|---|
| | $F_M$ | $F_G$ | M/G | $F_{MM}$ | $F_{GG}$ | $F_{MG, GM}$ | η | |
| *Laminaria ochroleuca* | 0.72 | 0.28 | 2.52 | 0.5 | 0.06 | 0.44 | 1.09 | [8] |
| *Laminaria brasiliensis* (blade) | 0.47 | 0.53 | 0.89 | 0.27 | 0.33 | 0.40 | 0.80 | [1] |
| *Laminaria brasiliensis* (stipe) | 0.46 | 0.54 | 0.85 | 0.23 | 0.31 | 0.46 | 0.93 | [1] |
| *Saccorhiza polyschides* (blade) | 0.42 | 0.58 | 0.72 | 0.3 | 0.46 | 0.24 | 0.49 | [1] |
| *Saccorhiza polyschides* (stipe) | 0.51 | 0.49 | 1.04 | 0.36 | 0.34 | 0.30 | 0.60 | [1] |
| *Saccharina japonica* | 0.65 | 0.35 | 1.85 | 0.48 | 0.18 | 0.34 | 0.75 | [33] |
| *Saccharina latissima* (old blade) | 0.5 | 0.5 | 1.01 | 0.36 | 0.35 | 0.30 | 0.60 | [34] |
| *Saccharina latissima* (old stipe) | 0.56 | 0.44 | 1.27 | 0.39 | 0.27 | 0.34 | 0.69 | [34] |
| *Saccharina longicruris* | 0.41 | 0.59 | 0.69 | 0.07 | 0.25 | 0.68 | 1.41 | [35] |
| *Lessonia nigrescens* | 0.59 | 0.41 | 1.43 | 0.4 | 0.22 | 0.38 | 0.79 | [30] |
| *Lessonia trabeculata* (blade) | 0.38 | 0.62 | 0.61 | 0.21 | 0.47 | 0.30 | 0.64 | [9] |
| *Ecklonia arborea* | 0.52 | 0.48 | 1.08 | 0.37 | 0.33 | 0.30 | 0.60 | [32] |
| *Ecklonia maxima* | 0.55 | 0.45 | 1.22 | 0.32 | 0.22 | 0.64 | 1.29 | [1] |
| *Durvillaea potatorum* | 0.68 | 0.32 | 2.125 | 0.56 | 0.2 | 0.24 | 0.55 | [31] |
| *Durvillaea willana* | 0.72 | 0.28 | 2.57 | 0.51 | 0.07 | 0.42 | 1.04 | [26] |
| *Durvillaea antarctica* | 0.68 | 0.32 | 2.15 | 0.51 | 0.16 | 0.34 | 0.78 | [30] |
| *Marginariella boryana* | 0.44 | 0.56 | 0.79 | 0.28 | 0.4 | 0.32 | 0.65 | [26] |
| *Ascophyllum nodosum* | 0.61 | 0.39 | 1.56 | 0.46 | 0.23 | 0.32 | 0.67 | [28] |
| *Saccorhiza polyschides* | 0.63 | 0.37 | 1.73 | 0.38 | 0.11 | 0.50 | 1.09 | [8] |
| *Laminaria ochroleuca* (blade) | 0.47 | 0.53 | 0.89 | 0.09 | 0.16 | 0.75 | 1.51 | This study |
| *Laminaria ochroleuca* (stipe) | 0.58 | 0.42 | 1.38 | 0.24 | 0.08 | 0.68 | 1.39 | |
| *Saccorhiza polyschides* (blade) | 0.62 | 0.38 | 1.62 | 0.37 | 0.14 | 0.49 | 1.04 | |
| *Saccorhiza polyschides* (stipe) | 0.68 | 0.32 | 2.14 | 0.44 | 0.08 | 0.48 | 1.10 | |
| Commercial alginate *(Sigma-Aldrich Na-Alginate)* | 0.77 | 0.23 | 3.42 | 0.68 | 0.14 | 0.18 | 0.51 | |

### 3.4. Viscosity and Molecular Weight

Viscosity is considered among the most important physical properties used to assess the gelling capability of alginates [1,5]. The studied species exhibited low intrinsic viscosity values ranging from 1.14 dL/g to 2.84 dL/g (Table 4), reflecting the low G-block content in their alginates block structures. Differences were found between species as alginates extracted from *S. polyschides* showed a very low viscosity compared to alginates from *L. ochroleuca*. This difference was also manifested in the tissue type of each species, and was particularly remarkable in *Laminaria* blades, showing the relative scarceness of MM blocks in its alginate structure. In fact, the intrinsic viscosity of alginates from the blade and stipe of *S. polyschides* were 1.56 dL/g and 1.14 dL/g, respectively, whereas viscosity found in *L. ochroleuca* blades (2.84 dL/g) exhibited a notably higher value than that found in its stipe (1.83 dL/g). The subsequent results of their molecular weight are shown in Table 4. A very low Mw was found in the stipe and blade of *S. polyschides*, indicating $0.53 \times 10^{-5}$ g/mol and $0.73 \times 10^{-5}$ g/mol, respectively. A similar result was noted from the stipe of *L. ochroleuca* (Mw = $0.81 \times 10^{-5}$ g/mol), unlike the prominent Mw value of its blade ($13.37 \times 10^{-5}$ g/mol). However, all results remain remarkably low compared with those found in the literature. A comparison of the alginates block structure suggests that the calculated Mw, however, correlates well with the composition of the so-called low-M

and intermediate alginates extracted from both species. This disparity in the monomer frequencies, sequential structure, viscosity, and molecular weight of the polymeric chain depends on many factors. Several studies have reported the relationship between the source of alginates and their composition. The harvesting location and seasonality of the extracted seaweeds were also correlated with the alginates block structure [36]. The latter was also proved to be strongly affected by the age and tissue type of the seaweeds, in addition to the environmental conditions of their habitat [9,34,37]. According to Draget et al. [1], alginates extracted from *S. polyschides* displayed a lower M/G ratio compared to that found in alginates from the Moroccan *S. polyschides* used in this study (M/G ratio = 1.62). Hence, this variation might be related to the sampling site of both seaweeds. However, a different structure was revealed in alginates extracted from the blade of both kelps, which were collected from the exact same sampling location as that of the present study [8]. Thus, the composition of the extracted alginates might not be affected by the aforementioned aspect. In fact, as reported by Belattmania et al. [8], the alginates extracted from the blades of *S. polyschides* and *L. ochroleuca* showed substantial differences in their M/G ratio, the values of which are 1.73 and 2.52, respectively, whereas results of the present study exhibited a lower M/G ratio in the blades of both species (M/G = 1.62 for *S. polyschides* and M/G = 0.89 for *L. ochroleuca*). According to earlier experience with *Laminaria* species [16,38] and *S. polyschides* [6], the chemical content of their blades is expected to undergo marked seasonal variations, which might be the reason behind the obtained differences in the alginate composition. Hence, along with the seasonality, the variation found in this study could be related to other parameters such as the age and thalli parts, also known to exert a considerable influence on alginates. According to Haug et al. [39], the older parts of seaweeds are the main reason behind the variation in alginates composition between different species, since the proportion of guluronic acid increases as the tissue grows older. This aspect was not taken into consideration while choosing the algal biomass used in the extraction, although the relatively high MM blocks content found in the examined alginates could be closely tied to an unintentional use of younger specimens. According to Venegas et al. [9] and Craigie et al. [37], alginates' composition might be influenced by the tissue type. In general terms, the stipe is most enriched in guluronic acid, as reflected by its usual relatively low M/G [24]. However, the results of this study do not support these findings, since the estimated M/G of the stipes (Table 3) were higher than those found in the blade, referring to the high mannuronate content in their alginates composition which imparts a greater flexibility to their algal tissue [19]. Indeed, the biological significance of alginate in the pheophyceae is generally believed to be that of a structure-forming component. In the present study, both kelps colonizing subtidal pools need to be physically flexible to survive the turbulent waters in their habitat, and withstand the wave action on their tissues, which also increases with strong currents. Thus, these variations in the alginate's composition might be explained by the hydrodynamic environment of the local *L. ochroleuca* and *S. polyschides*, since they both share the same dwelling. This natural variability in alginates provides a wide range of functional properties allowing several applications that determine their commercial value [32,40]. Essentially, alginates with a high M/G ratio would yield more elastic gels, desirable for food, cosmetic, or pharmaceutical products, whereas alginates with a low M/G are more suitable for cell encapsulation and biomedical or environmental applications due to the resulting strong and brittle gels [41].

**Table 4.** Intrinsic viscosity and molecular weight of the extracted alginates compared to data from various kelp species.

| Kelp Species | $[\mu]$ (dL/g) | $M_W \times 10^{-5}$ (g/mol) | References |
|---|---|---|---|
| *Macrocystis pyrifera* | 9.43 | 0.76 | [42] |
| *Laminaria hyperborea* (blade) | 5 | 1.55 | [43] |
| *Laminaria hyperborea* (stipe) | 5.2 | 1.6 | [43] |

**Table 4.** *Cont.*

| Kelp Species | [μ] (dL/g) | $M_W \times 10^{-5}$ (g/mol) | References |
|---|---|---|---|
| *Ascophyllum nodosum* | 2.8 | 1.32 | [44] |
| *Laminaria japonica* | 15.4 | 7.44 | [44] |
| *Laminaria ochroleuca* (blade) | 2.84 | 1.34 | |
| *Laminaria ochroleuca* (stipe) | 1.43 | 0.66 | |
| *Saccorhiza polyschides* (blade) | 1.56 | 0.73 | This study |
| *Saccorhiza polyschides* (stipe) | 1.14 | 0.53 | |
| Commercial alginate *(Sigma-Aldrich Na-Alginate)* | 1.83 | 0.85 | |

## 4. Conclusions

The present study aims to characterize the alginates extracted from the two parts (stipe and blade) of the kelps *L. ochroleuca* and *S. polyschides*. Yields showed interesting values ranging from $20.19 \pm 2$ to $49.8 \pm 2.4\%$ dw, which correspond to the average of most of kelp species used in the industrial production of alginates. The highest alginate contents were found in the stipe of both species. FT-IR spectroscopy exhibited similar spectra between the alginates extracted from both kelps and the commercial alginate. However, [1]H-NMR results revealed a disparity in the structural composition of the extracted alginates with M/G values ranging from 0.89 to 2.14. This variation was also shown through the viscosity and Mw indicating very low values. High mannuronate content was found in all alginates except those extracted from the blade of *L. ochroleuca*. The quasi-dominance of mannuronic ($F_{MM}$) and guluronic ($F_{GG}$) homopolymers over heteropolymeric fractions ($F_{GM}$ and $F_{MG}$) was detected in *S. polyschides*. Nevertheless, the heteropolymeric sequences dominated the diads of the alginates extracted from *L. ochroleuca*. Thus, with the optimization of extraction using different parts of the thalli, both *L. ochroleuca* and *S. polyschides* could be considered as potential alginophytes. However, in order to preserve these kelps species, seaweed farming would be a better alternative to guarantee a sufficient biomass for industrial extraction, and establish the local sustainability of the species known for their prominent ecological role.

**Author Contributions:** Conceptualization, B.S., F.B. and A.R.; methodology, S.K., Z.B., B.S. and A.R.; software, C.J. and F.B.; validation, C.J., F.B. and B.S.; formal analysis, C.J., F.B. and S.K.; writing—original draft preparation, S.K.; writing—review and editing, B.S. and Z.B.; visualization, C.J., F.B., S.K., K.K. and Z.B.; supervision, B.S. and A.R.; project administration, B.S. and C.J.; funding acquisition, B.S. and C.J. All authors have read and agreed to the published version of the manuscript.

**Funding:** This research was funded by the project VPMA3/DESRS-ANPMA-CNRST 'Exploitation de la diversité spécifique et génétique pour une bioraffinerie innovante des algues marines de la côte atlantique marocaine' and also supported by the FCT Portugal—CNRST Morocco bilateral cooperation project 'Variation of marine-forests traits in range-edge vs. core NE Atlantic upwelling refugia in a context of climatic change'.

**Informed Consent Statement:** Not applicable.

**Data Availability Statement:** All data are reported within this manuscript.

**Acknowledgments:** S.K. acknowledges her Excellence doctoral grant n° 5UCD2017 from The National Centre for Scientific and Technical Research (CNRST, Morocco) and the mobility grant within the Deep Blue Project: Developing Education and Employment Partnerships for a Sustainable Blue Growth in the Western Mediterranean Region.

**Conflicts of Interest:** The authors declare no conflict of interest. The funders had no role in the design of the study; in the collection, analyses, or interpretation of data; in the writing of the manuscript; nor in the decision to publish the results.

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
