# Peer review of "Isolation and Structural Characterization of Alginates from the Kelp Species Laminaria ochroleuca and Saccorhiza polyschides from the Atlantic Coast of Morocco"

_colloids, doi:10.3390/colloids6040051_

Round 1
Reviewer 1 Report
The manuscript is about isolation, structural characterization and rheological proprieties of alginates obtained from different species and tissues of seaweeds.
This article gives some new results on the differences between alginates from blades and stipes. But unfortunately, the novelty and purpose of this study is not quite clear. I suppose these points should be covered in the abstract and introduction. The respected authors also should explain how the specific results will be useful for the field.
The study of rheological properties is limited only by the study of intrinsic viscosity. At the same time, the authors mention gel-forming properties in the abstract. In this case, a more detailed and deep study of rheological properties should be carried out. For example, dynamic analysis of viscoelastic properties , the determination of the time and temperature of gelation by analyzing the evolution of dynamic modules (G’ and G’’), etc.
Author Response
Reviewer 1
The manuscript is about isolation, structural characterization and rheological proprieties of alginates obtained from different species and tissues of seaweeds.
This article gives some new results on the differences between alginates from blades and stipes. But unfortunately, the novelty and purpose of this study is not quite clear. I suppose these points should be covered in the abstract and introduction. The respected authors also should explain how the specific results will be useful for the field.
The study of rheological properties is limited only by the study of intrinsic viscosity. At the same time, the authors mention gel-forming properties in the abstract. In this case, a more detailed and deep study of rheological properties should be carried out. For example, dynamic analysis of viscoelastic properties the determination of the time and temperature of gelation by analyzing the evolution of dynamic modules (G’ and G’’), etc.
Response
Dear Reviewer,
Thank you for your relevant revision.
The authors are tried to highlight the novelty of this study. In fact, the exploitation of the stipe and the blade would improve the yield of the extracted sodium alginates while providing more chemical structures that may expand the hydrocolloid applications, mainly in biotechnology. This information was added in the abstract and the introduction to show the novelty and purpose of this study. The obtained sodium alginates would be more suitable for the biotechnology fields considering their chemical structure and viscosity.
Regarding the rheological properties, indeed, this was changed with viscosity to meet with the results proposed in the manuscript.
Please you find attached the revised version of the ms.

Reviewer 2 Report
The article is devoted to the analysis of the composition and physicochemical properties of the alginate obtained from different thalli parts (blade and stipe) of brown algae (Laminaria ochroleuca and Saccorhiza polyschides) collected on the Moroccan coast. Marine-derived polysaccharides, in particular sodium alginates, obtained from renewable sources, are currently very popular biopolymers. They are widely used in food technology, in the production of cosmetics and medicines. Expanding the list of raw materials for obtaining this polysaccharide is an urgent task.
Using various physicochemical methods, including IR spectroscopy, NMR and viscometry, the authors showed that the extracted alginates are not inferior to commercial samples in terms of component composition and molecular weight. The presented article is written in good scientific language, logically structured, the statements are confirmed by experimental results. The presented results will be of interest to the readers of the journal, especially those who are interested in polysaccharides, their properties and practical use.
There is a small remark. It is necessary to correct typos: the values ​​of the molecular weight of the polysaccharide (lines 280-282); for example, the molecular weight is 0.53×105 g/mol, and not 0.53×10-5 g/mol.
I believe that the article can be printed in the presented version (after correcting typos).
Author Response
Reviewer 2
The article is devoted to the analysis of the composition and physicochemical properties of the alginate obtained from different thalli parts (blade and stipe) of brown algae (Laminaria ochroleuca and Saccorhiza polyschides) collected on the Moroccan coast. Marine-derived polysaccharides, in particular sodium alginates, obtained from renewable sources, are currently very popular biopolymers. They are widely used in food technology, in the production of cosmetics and medicines. Expanding the list of raw materials for obtaining this polysaccharide is an urgent task.
Using various physicochemical methods, including IR spectroscopy, NMR and viscometry, the authors showed that the extracted alginates are not inferior to commercial samples in terms of component composition and molecular weight. The presented article is written in good scientific language, logically structured, the statements are confirmed by experimental results. The presented results will be of interest to the readers of the journal, especially those who are interested in polysaccharides, their properties and practical use.
There is a small remark. It is necessary to correct typos: the values of the molecular weight of the polysaccharide (lines 280-282); for example, the molecular weight is 0.53×105 g/mol, and not 0.53×10-5 g/mol.
I believe that the article can be printed in the presented version (after correcting typos).
Response
Dear Reviewer,
The authors are grateful for your relevant remarks, Mw×10-5 ist he correct typing. The Mwwas estimated by the Mark–Houwink–Sakurada equationusingthe empirical relations for [η]and the weight-average molar mass (Mw), where [η] is given in dL/g and M in kDaltons.
The equivalent of 1kDaltons is 10-3 g/mol, and following the literature and the obtained values, results were given in Mw× 10-5
Please you find attached the revised version of the ms.

Reviewer 3 Report
Dear Authors,
During analysis of algiantes you use commercial sodium alginate as control but in the last part of study (viscosity) ther is no control. Would you plaese add the viscosity of commercial sodium alginate. How many repetitions were made during each study?
Author Response
Reviewer 3
Dear Authors,
During analysis of algiantes you use commercial sodium alginate as control but in the last part of study (viscosity) ther is no control. Would you plaese add the viscosity of commercial sodium alginate. How many repetitions were made during each study?
Response
Dear Reviewer, thank you for your comments. The viscosity of the commercial alginate is indicated in the table 4 under its reference name “Sigma-Aldrich Na-Alginate”. An amendment to tables 3 and 4 has been made to specify that the former corresponds to the commercial alginate used as a control. This information has also been added to the text where it is mentioned in lines 164 and 349.
3 repetitions were made for each study.

Reviewer 4 Report
Dear Authors,
your manuscript presents valuable experimental investigations about extracted alginate in order to get their structural features. You measured also the viscosity and deduced from it the molecular weight. However, several revisions, many of them being major, are needed. In addition, some questions should be answered before resubmission. Please find below my recommendations and questions which I hope can assist you in your revision:
1) Please correct the title “properties” and not “proprieties”. Also your paper does not contain rheological measurement (you have only measured viscosity to deduce the Molecular weight) so please delate rheological properties in the title.
2) Please in line 134 (page 4) write the equation correctly so one could see that “a” is an exponent: (Mw)a. Then specify that a is temperature dependent.
3) Replace “net-work” by simply “network” for example in page 7.
4) Could you please give a convincing argument why you used a=0.984 for the exponent of the Mark–Houwink–Sakurada equation, because the temperature of the experiment and more generally the conditions can be different than the reference (13) from which you took this value. In addition, the low molecular weight value could be due to this choice of the “a” exponent.
Best Regards
Author Response
Reviewer 4
Dear Authors,
your manuscript presents valuable experimental investigations about extracted alginate in order to get their structural features. You measured also the viscosity and deduced from it the molecular weight. However, several revisions, many of them being major, are needed. In addition, some questions should be answered before resubmission. Please find below my recommendations and questions which I hope can assist you in your revision:
1) Please correct the title “properties” and not “proprieties”. Also your paper does not contain rheological measurement (you have only measured viscosity to deduce the Molecular weight) so please delate rheological properties in the title.
2) Please in line 134 (page 4) write the equation correctly so one could see that “a” is an exponent: (Mw)a. Then specify that a is temperature dependent.
3) Replace “net-work” by simply “network” for example in page 7.
4) Could you please give a convincing argument why you used a=0.984 for the exponent of the Mark–Houwink–Sakurada equation, because the temperature of the experiment and more generally the conditions can be different than the reference (13) from which you took this value. In addition, the low molecular weight value could be due to this choice of the “a” exponent.
Best Regards
Response
Dear reviewer,
We are grateful for your relevant remarks and suggestion. We have tried to response for each point you indicated:
1- Typos errors were corrected and the title was modified as:
2- The equation was rectified using the correct typing of (Mw)a.
3- As requested “net-work” was replaced by “network”
4- The choice of the empirical relations for [g] and the weight-average molar mass (Mw) depends on the methodology adopted, in particular the temperature conditions when evaluating the viscosity, which in the case of the Clementi et al. were similar to the temperature adopted in our experiment, i.e. 25°C
Clementi et al. proposed two empirical relations stating that “there was no statistically significant difference between them”, although we have opted for the one proposing a=0.984 and k=0.023 asconstants.
That equation with a relatively higher r2 (r2=0.967) was better adapted to macroalgae such as Fucus, Laminaria compared to the other equation which was relatively more adequate to bacterial alginate.
The selected empirical relation proposing a= 0.984 was subsequently approved and adopted by several studies such as Torres et al. 2007 and Chee et al.2010. The latter were also used as a reference for the methodology adopted in the present study, since they focus on the explorationalgal alginate using similar temperature.
Based on these facts, the empirical relationship adopted in the present study was that using a=0.984 for the exponent of the Mark–Houwink–Sakurada equation
In order to remove this ambiguity, the reference of Clementi et al. 1998 was replaced by Torres et al.2007.
Please you find attached the revised version of the ms.

Round 2
Reviewer 1 Report
The Manuscript has been improved, so I recommend to accept the MS in its curent form.
Reviewer 4 Report
Dear Authors,
Your Manuscript has been well revision and the pointed out remarks were addressed. So congratulations for your nice work which I recommended its acceptance as it is.
Best Regards